# Confronting the Boundary Layer Data Gap: Evaluating New and Existing Methodologies of Probing the Lower Atmosphere

Tyler M. Bell[1,2], Brian R. Greene[1,2,3], Petra M. Klein[1,2], Matthew Carney[1], and Phillip B. Chilson[1,2,3]

[1]University of Oklahoma School of Meteorology, Norman, Oklahoma
[2]University of Oklahoma Center for Autonomous Sensing and Sampling, Norman, Oklahoma
[3]University of Oklahoma Advanced Radar Research Center, Norman, Oklahoma

**Correspondence:** Tyler Bell (tyler.bell@ou.edu)

**Abstract.** It is widely accepted that the atmospheric boundary layer is drastically under-sampled in the vertical dimension. In recent years, the commercial availability of ground based remote sensors combined with the widespread use of small, weather-sensing uncrewed aerial systems (WxUAS) has opened up many opportunities to fill this measurement gap. In July 2018, the University of Oklahoma (OU) deployed a state-of-the-art WxUAS dubbed the CopterSonde, as well as the Collaborative Lower Atmospheric Mobile Profiling System (CLAMPS), in the San Luis Valley in south-central Colorado. Additionally, these systems were deployed to the Kessler Atmospheric and Ecological Field Station (KAEFS) in October 2018. The co-location of these various systems provided ample opportunity to compare and contrast kinematic and thermodynamic observations from different methodologies of boundary layer profiling, namely: WxUAS, remote sensing, and the traditional in-situ radiosonde. In this study, temperature, dew point temperature, wind speed, and wind direction from these platforms are compared statistically with data from the two campaigns. Moreover, we present select instances from the data set to highlight differences between the measurement techniques. This analysis highlights strengths and weaknesses of PBL profiling and helps lay the groundwork for developing highly adaptable systems that integrate remote and in-situ profiling techniques.

## 1 Introduction

It has been a decade since the National Research Council (2009) published their report detailing the need for a mesoscale observation network that extends vertically beyond the Earth's surface. It was concluded that vertical profiles in the boundary layer are inadequately measured in both space and time. At the time, the only operational method of resolving the thermodynamic and kinematic properties of the boundary layer was with radiosondes launched twice daily by weather services around the world, which has been conducted since the 1930s. However, radiosondes spend only a limited amount of time in the boundary layer every day. Due to the importance of the boundary layer characteristics to areas like air quality forecasts (World Health Organization, 2016; Park et al., 2016; Bessagnet et al., 2019), convective initiation (e.g., Nowotarski et al., 2011; Markowski and Bryan, 2016; Markowski, 2016; Koch et al., 2018), and wind energy forecasting (e.g., Zhou and Chow, 2012), more observations are needed to fill the spatial and temporal data gap that exists to date. The recent Decadal Survey by the National Academy of Sciences (National Academies of Sciences and Medicine, 2018) further reinforces this by identifying boundary layer processes as key to improving weather and climate models. To fill this data gap, two different paradigms have been

suggested: ground-based remote sensing and uncrewed aircraft systems (UAS, also referred to as remotely piloted aircraft systems, RPAS). Both paradigms have particular strengths and weaknesses that need to be characterized before a system can be proposed for widespread adaptation.

A workshop was conducted shortly after the NRC report was published that identified several possible ground-based remote sensing platforms that could augment current upper-air and satellite observations (Hoff and Hardesty, 2012). Attendees generally agreed that instruments needed to have accuracy within $1°C$ for temperature and $1$ g kg$^{-1}$ for moisture while having a vertical resolution of 30–100 m. Additionally, the NRC report reccomended a temporal resolution ranging from 15 minutes to 3 hours, depending on the phenomena (National Research Council, 2009). At the time, three platforms largely met these requirements: microwave radiometers (MWR), Atmospheric Emitted Radiance Interferometers (AERI), and water vapor differential absorption lidars (WV-DIAL). Of these, the MWR and AERI stood out due to their ability to retrieve both temperature and moisture profiles, along with several other variables. Both instruments measure downwelling radiation emitted by the atmosphere. The measured bands encompass spectral regions where gasses have absorption features. These observations can be fed into multiple types of retrievals to obtain thermodynamic profiles of the atmosphere. While the WV-DIAL can provide higher resolution measurements, at the time of writing, there are still no commercially available WV-DIAL systems.

However, these remote sensing systems are expensive and multiple instruments are required to achieve both thermodynamic and kinematic profiles of the boundary layer. Recent workshops have identified UAS as a potential way to get high resolution thermodynamic and kinematic profiles of the boundary layer. For example, the National Center for Atmospheric Research Earth Observing Laboratory held a workshop focused on the use of UAS in atmospheric research (Vömel et al., 2018). They identified various areas in which improvements are needed for weather sensing UAS to be put to wide-spread use, with one of the most important being the need for a standard sensor packages for the various types of systems.

Due to regulatory issues related to integration of UAS into the National Airspace System (NAS), UAS are currently less developed and less utilized than remote sensors (Hoff and Hardesty, 2012; Koch et al., 2018), though the use of UAS is expected to continue to increase as further advances in GPS and communication allow for safer operation in public airspace (Geerts et al., 2018). Additionally, as commercial applications of UAS continue to emerge (e.g., remote package deliveries, infrastructure inspection, and agricultural monitoring) and safety cases are developed, the Federal Aviation Administration will have to plan and account for UAS in the NAS.

Traditionally, observations of the atmosphere using weather-sensing UAS (WxUAS) utilized fixed-wing aircraft (e.g., Reuder et al., 2009; Chilson et al., 2009; Houston et al., 2012; Reuder et al., 2012; Bonin et al., 2012; Balsley et al., 2013; Lawrence and Balsley, 2013; Lothon et al., 2014; Wildmann et al., 2014, 2015; Båserud et al., 2016; de Boer et al., 2016). Using fixed-wing WxUAS allowed researchers to leverage major technological advances from decades of measurement research from piloted aircraft (e.g., Saïd et al., 2005; Gioli et al., 2006; van den Kroonenberg et al., 2012). Additionally, fixed-wing WxUAS are capable of flight times of an hour or more, allowing researchers to cover a large spatial range.

However, the recent commercial boom of UAS has made rotary-wing UAS (rwUAS) easily accessible to the public. Researchers are now turning to rwUAS because they are more versatile, readily available, and are relatively easy to operate. A common application of rwUAS in atmospheric science is collecting thermodynamic and kinematic variables as a function of

altitude, similar to the traditional radiosonde. Forecasters and researchers can use the same conceptual framework as radiosondes to analyze and interpret data from rwUAS profiles. In a sense, the framework is even more applicable since rwUAS can remain over the same geographical location during observations while radiosondes drift many kilometers downwind.

Using rwUAS poses new challenges that must be overcome before they can be considered a viable platform for atmospheric observation. For example, rwUAS modify the environment surrounding them, so measuring even simple variables like temperature can be difficult. Careful considerations must be made to ensure the true environmental temperature is being measured, as opposed to the modified rwUAS environment. Additionally, proper aspiration of the sensors and shielding from solar radiation is vital to accurate measurements (Tanner et al., 1996; Hubbard et al., 2004; Greene et al., 2018, 2019).

In this study, a comparison is made between results from a rotary-wing WxUAS and different ground based remote sensors against data from the historical standard: the radiosonde. Observations of the temperature, dew point temperature, wind speed, and wind direction of the lower atmosphere were collected during two field campaigns under different environmental conditions with the instruments operated at the same time and from the same location. This provided an opportunity to both compare results from various instruments and to consider the strengths and weakness of the methodologies considered. Though comparisons between radiosondes and the ground-based remote sensors used in these campaigns are not novel (Pearson et al., 2009; Turner and Löhnert, 2014; Blumberg et al., 2015; Päschke et al., 2015; Koch et al., 2018; Turner and Blumberg, 2018), it is important to perform these comparisons for this study to assist in identifying and characterizing the uncertainty in all the systems. As mentioned before, previous studies have worked to mitigate the effect of the UAS on the thermodynamic measurements, which decouples the UAS technique from the sensor themselves (Greene et al., 2018, 2019). However these studies were performed in relatively idealized conditions. The purpose of this study is to verify that these results hold in a more "operational" mode. In addition to the comparisons, synergies between the various systems are identified and discussed. The following sections contain more information about the instrument systems and the field campaigns, an analysis of the results, and then a summary of the findings.

## 2 Observing systems

For this study, the three profiling platforms that are compared consist of the Collaborative Lower Atmospheric Mobile Profiling System (CLAMPS), the CopterSonde, and radiosondes. Each platform is described in detail below.

### 2.1 CLAMPS

CLAMPS consists of an AERI (Knuteson et al., 2004a, b), a Version 4 Humidity and Temperature Profiling MWR (HATPRO; Rose et al., 2005), and the Halo Photonics Streamline scanning Doppler lidar (DL; Pearson et al., 2009) in a modified off-the-shelf trailer (Fig. 1a). The CLAMPS facility also has the ability to launch radiosondes as there is storage for multiple helium tanks and an antenna mounted on top of the trailer (Wagner et al., 2019). The DL scan strategy consisted of an 18-point velocity azimuth display at 70 degrees elevation (VAD; Browning and Wexler, 1968; Päschke et al., 2015) and a 6-point VAD at 45 degrees elevation every 5 minutes to gather profiles of wind speed and direction. For the following analysis, only the 18-point

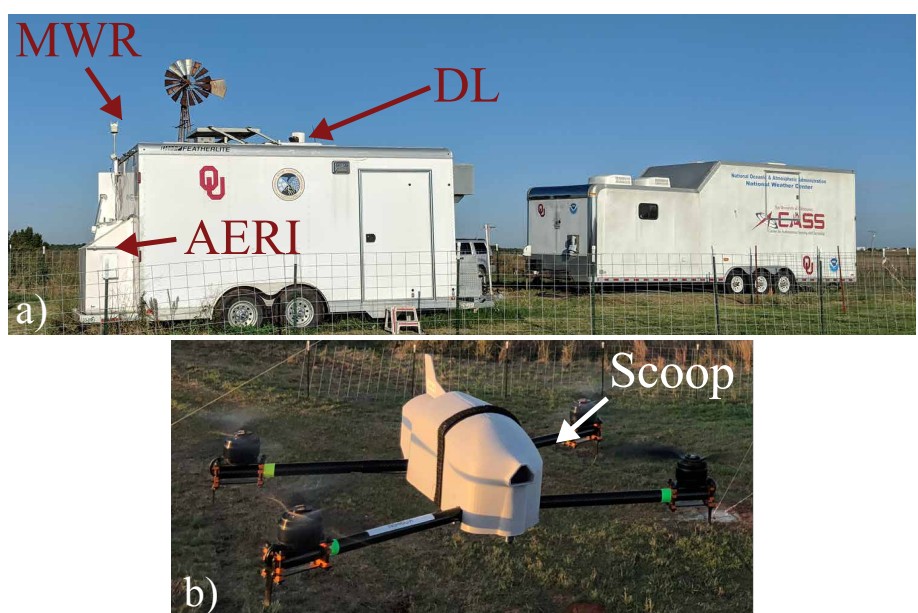

**Figure 1.** Photos of the CLAMPS facility (a, left) and the CopterSonde 2.5 (b) deployed at KAEFS.

VAD scan is used. When the DL was not performing these VAD scans, it was staring vertically to collect vertical velocity statistics.

Thermodynamic variables were collected using the AERI and MWR in a joint retrieval. The AERI Optimal Estimation (AERIoe; Turner and Löhnert, 2014; Turner and Blumberg, 2018) is a physical retrieval that retrieves temperature and water vapor profiles using an iterative optimal estimation technique (Rodgers, 2000). For the retrievals in this study, surface data from a Vaisala WXT-530 weather station mounted 3 m above ground level (AGL) on the back of CLAMPS was used to constrain the temperature and moisture at the surface. Additionally, colocated radiosondes were used to constrain the upper atmosphere above 3 km AGL since nearly all the information contained in the AERI measurement is contained in the lowest 2 km of the atmosphere (Turner and Löhnert, 2014). The backscatter from the DL vertical stare was used to detect cloud base in the retrieval.

## 2.2 CopterSonde 2.5

The WxUAS used by OU is the CopterSonde v2.5 (hereafter, just CopterSonde). The CopterSonde is the second iteration of the rwUAS that is described in detail in Greene et al. (2018). The original CopterSonde was built and deployed by the Center for Autonomous Sensing and Sampling (CASS) at OU for the Environmental Profiling and Initiation of Convection (EPIC; Koch et al., 2018) field campaign. CASS originally planned to use an off-the-shelf rwUAS, but it was determined that an airframe was not available to account for the specific needs of environmental sampling and instead opted to build a custom WxUAS.

The original CopterSonde was successfully built and deployed for the EPIC campaign, but found mixed success in providing accurate and reliable atmospheric thermodynamic data (Koch et al., 2018).

Deployment of the original CopterSonde provided many valuable lessons on how to improve upon its design and mode of operation. This led to the development of the CopterSonde for deployment in the 2018 Innovative Strategies for Observations in the Arctic Atmospheric Boundary Layer (ISOBAR) campaign (Kral et al., 2018). The CopterSonde is a quad-copter designed specifically for thermodynamic and kinematic profiling. Instead of temperature and humidity sensors being "passengers" on the CopterSonde, they are directly integrated into the nose of the craft utilizing a custom 3D printed shell and are read by custom autopilot code. Additionally, the CopterSonde is programmed to always turn into the wind. This combined with radiation shielding and a ducted fan to aspirate the sensors increases the precision of the measurements from the platform, which minimizes the effect of the platform itself on the sensors (Greene et al., 2019). More details of the development of the CopterSonde can be found in Segales et al. (2019, in review).

Data from the CopterSonde are processed to a 3 m vertical resolution starting from 6 m AGL in order to not contaminate the profile with effects induced by the ground. The temperature measurements were made using three iMet-XF glass bead thermistors while the relative humidity measurements were made using three Innovative Sensor Technology HYT 271 relative humidity sensors. Three sensors were used for redundancy; if one sensor goes bad, the other two can be used to automatically identify the bad sensor. If only 2 sensors were used and a sensor malfunctioned, it would be difficult to determine the faulty sensor automatically. Wind speed and direction are calculated by using a methodology based on Neumann and Bartholmai (2015), which uses the tilt of the airframe to estimate the velocity. This is done in real time so that custom autopilot software can always direct the nose of the CopterSonde into the wind, which improves thermodynamic and kinematic measurements (Greene et al., 2019). Herein, the thermodynamic package and ducted fan are referred to as the "scoop".

The sensors in the CopterSonde scoop were characterized in the Oklahoma Climatological Survey calibration laboratory. The entire scoop was placed inside of a controlled calibration chamber and aspirated using the ducted fan to account for any heat that may come off of the fan. The scoop was calibrated for 1-hr periods at multiple chamber reference temperatures and humidities. Data from each sensor in the scoop were compared to a calibrated thermistor and a chilled-mirror hygrometer and any sensor bias was corrected. CopterSonde data used in this analysis have had these correction curves applied.

## 2.3 Radiosondes

The Vaisala RS92-SGP was the radiosonde used for this study. The data used in the comparisons was processed using the Vaisala ground station software. The targeted ascent rate was 5 m s$^{-1}$ with the radiosonde reporting to the ground station every second. The files used for the analysis were post processed to 2 second resolution. The Vaisala RS92-SGP has a temperature uncertainty of 0.5°C, relative humidity uncertainty of 5%, wind speed uncertainty of 0.15 m s$^{-1}$ (at wind speeds > 3m s$^{-1}$), and wind direction uncertainty of 2 degrees according to the Vaisala specifications.

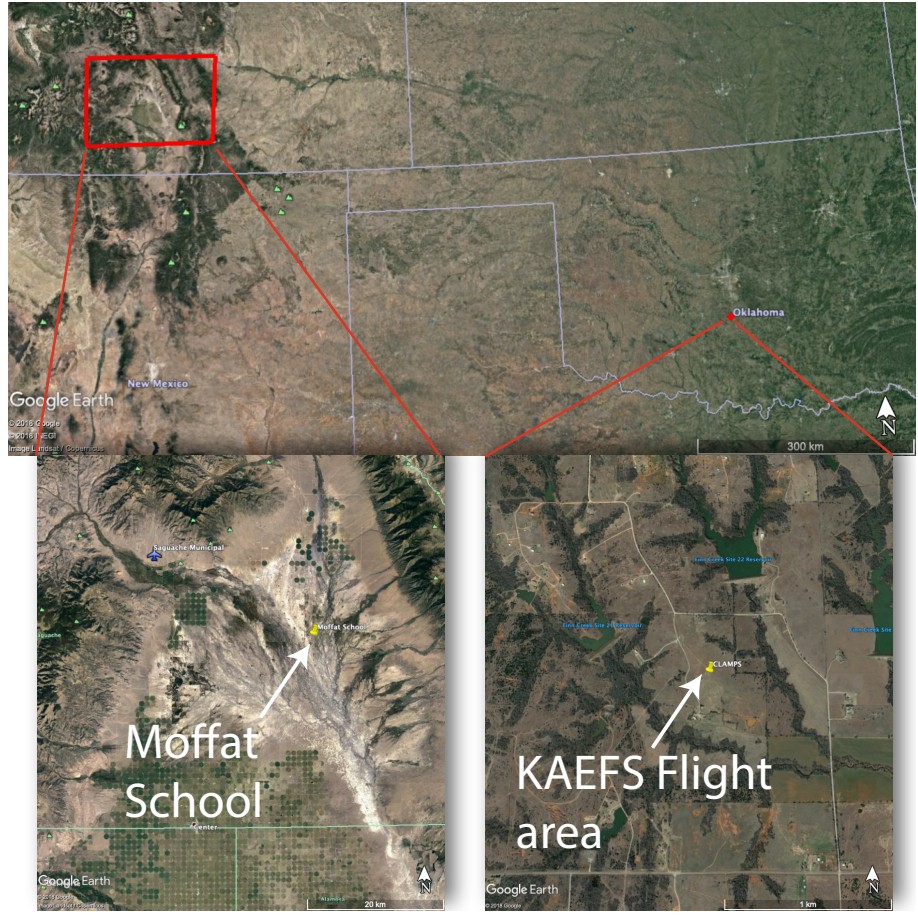

**Figure 2.** Location and domain size for LAPSE-RATE (a) and Flux Capacitor (b) campaigns. During LAPSE-RATE, OU was deployed at both the Saguatch Municipal Airport and the Moffat School, however only the Moffat data will be analyzed in this study.

## 3    Campaigns

The CopterSonde and CLAMPS were colocated for two field experiments in 2018: Lower Atmospheric Process Studies at Elevation - a Remotely-piloted Aircraft Team Experiment (LAPSE-RATE) and Flux Capacitor. These experiments together provide a wide range of conditions under which the CopterSonde and CLAMPS were evaluated. The experiments are described below.

### 3.1    LAPSE-RATE

In July 2018, OU deployed the CopterSonde and CLAMPS in the San Luis Valley in south-central Colorado (Fig. 2a) as part of Lower Atmospheric Process Studies at Elevation - a Remotely-piloted Aircraft Team Experiment (LAPSE-RATE, de Boer et al., 2019, accepted). LAPSE-RATE was a week long experiment with the goal of dispersing UAS throughout the valley

**Table 1.** Summary of flights at the Moffat School and KAEFS. In total, 141 flights were used in the comparison. Column 5, lists the typical cadence of flights, typical $\Delta t$, which corresponds to the typical amount of time between flights.

| Date | Number of Flights | First Flight (UTC) | Last Flight (UTC) | Typical $\Delta t$ (min) | Site |
|---|---|---|---|---|---|
| 20180715 | 18 | 1326 | 1944 | 15 | Moffat |
| 20180716 | 20 | 1515 | 2115 | 15 | Moffat |
| 20180717 | 17 | 1330 | 1633 | 15 | Moffat |
| 20180718 | 16 | 1231 | 1603 | 15 | Moffat |
| 20180719 | 24 | 1115 | 1700 | 15 | Moffat |
| 20181005 | 18 | 1500 | 2335 | 30 | KAEFS |
| 20181006 | 28 | 0000 | 1431 | 30 | KAEFS |

to study various atmospheric phenomena, including convective initiation, cold air drainage, and morning transitions. Broader goals were to further evaluate and advance UAS that have been developed for atmospheric sensing in recent years. LAPSE-RATE was organized in connection with the annual meeting of the International Society for Atmospheric Research using Remotely piloted Aircraft (ISARRA). Over 100 participants from 16 institutions took part in the campaign. The flight teams
logged almost 1,300 individual flights, with CASS contributing nearly 200 of those flights. During the week, one of the largest UAS intercomparison and calibration efforts to date also took place, highlighting some of the thermodynamic and kinematic sampling biases inherent to different varieties of aircraft (Barbieri et al., 2019).

The San Luis valley provided a unique opportunity to study flow in complex terrain, due to it being surrounded by mountains to the east and west. The valley floor is primarily flat shrubland and irrigated agricultural fields (Fig. 2a). The mountains create
complex thermal flows while the irrigated fields on the west half of the valley, contrasted with the shrubland on the east half, create complex near-surface moisture gradients.

OU primarily operated at two sites during the campaign, the Moffat Consolidated School in the center of the valley and the Saguache Municipal Airport, which was 25 km to the NW of the Moffat School. For one mission, the OU team at Saguache Municipal Airport relocated 10 km to the SE to capture drainage flow from the mountains. CLAMPS was located at the Moffat
school and CopterSondes were flown at both sites. Based on permission from the FAA, flights were conducted from the surface to 910 m AGL along a vertical trajectory. Three different CopterSondes were flown during the LAPSE-RATE campaign. Two of the CopterSondes were flown solely at the Moffat school. The third was flown at Saguatch and the drainage flow site mentioned above. Since all three scoops were characterized in the calibration chamber, there should be minimal difference in the measurements from each platform given that they are otherwise identical. Additionally, the CopterSonde was validated
against a 15 m mobile mast during the LAPSE-RATE campaign (Barbieri et al., 2019).

### 3.2 Flux Capacitor

The Flux Capacitor campaign was an internally funded experiment organized by various groups at OU. One of the main goals was to evaluate the performance of the CopterSonde over a full diurnal cycle in support of the 3D Mesonet concept (Chilson et al., 2019). The campaign took place at the Kessler Atmospheric and Ecological Field Station (KAEFS) in October 2018. KAEFS is located 28 km southwest of the OU Norman campus and has been the main test site for the CopterSonde. In addition, KAEFS facilitates instrumentation from multiple research groups at OU, including the Washington site of the Oklahoma Mesonet.

For Flux Capacitor, CLAMPS was again colocated with the launch site for the CopterSonde (Fig. 2b), allowing comparisons between the systems. CopterSonde flights were conducted on a 30-minute interval up to a max altitude of 1,200 m AGL. In total, 46 flights were conducted over a 24 hour period (Table 1). Two flights were missed due to mechanical issues with the CopterSonde, which have since been addressed.

Details of the number and times of flights used in this analysis (from both campaigns) are shown in Table 1.

## 4 Results

To evaluate the different instruments, a detailed statistical comparison of each system relative to the others is shown in Section 4.1 and a couple of case studies are shown in Section 4.2. The statistical comparison provides an overall picture of how well the systems perform relative to one another while the case studies identify specific issues that can appear in data from each system. Herein, statistical comparisons are calculated using the open-source python packages NumPy (van der Walt et al., 2011), SciPy (Virtanen et al., 2019), and scikit-learn (Pedregosa et al., 2011), and figures are visualized using Matplotlib (Hunter, 2007).

### 4.1 Statistical Comparison of Systems

It is important to characterize differences statistically to draw conclusions about system performance relative to each other. For each of the system combinations (CLAMPS-Copter, CLAMPS-Radiosonde, Copter-Radiosonde), data from all heights and from both campaigns are compared to one another. A large number of data points are needed to draw any conclusions since the measurements from each platform are based on different assumptions and affected by different sources of error. While all three profiling methods are most often interpreted as a perfect vertical profile, this is obviously not true. For example, the VAD scan assumes that the wind field is horizontally homogeneous and stationary in order to retrieve the horizontal wind vector. This means the DL observations are averaged spatially both in terms of radial bins and the PPI scan required to calculate the VAD. Additionally, radiosondes drift and can often be collecting in-situ measurements far from the launch point. So while comparing the systems one-to-one may not be ideal, at this point it is the best approach available to us.

Due to these limitations, any spread in the data presented from the following analyses contains three components: one component from instrument imprecision, one from the inability of the instruments to measure the same point at precisely

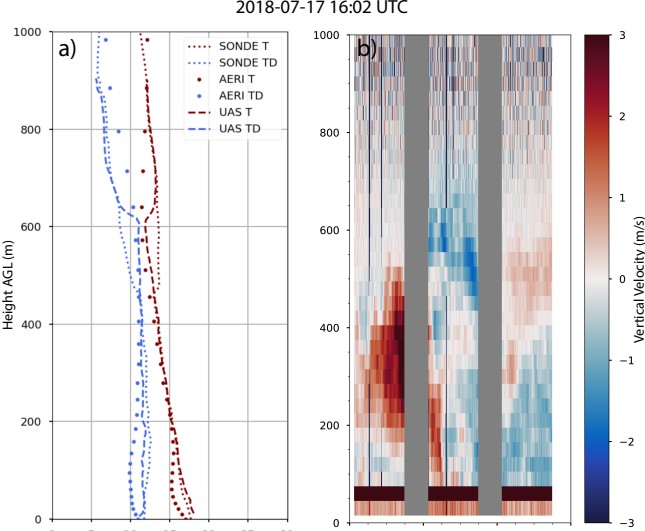

**Figure 3.** Example profile of temperature (red) and dew point temperature (blue) from each observation platform (a) and a time height cross section of vertical velocity measured from the CLAMPS DL from the same time period (b). The grey areas in (b) indicate the DL was not in vertical stare mode and was performing PPI scans.

the same time, and one component that arises from the differences in measurement technique. The goal of this study was to examine the spread that arises from the measurement technique itself.

Therefore, in order to eliminate any differences due to changing atmospheric conditions, data points with a difference that lie outside the $2\sigma$ envelope are considered outliers and were removed from the analysis. As a result, most of the comparisons had under 10 percent of the points removed. However, the kinematic comparison between the radiosonde and the DL had the largest percentage of points removed at 33 percent. This was due to a combination of fewer overall points and the VAD technique and radiosondes struggling to capture the very low wind speeds observed during the daytime hours of LAPSE-RATE. In general, approximately 15 percent of profiles had at least one outlier. Of these 15 percent, the median number of outliers per profile was 2. These outliers followed no apparent pattern and would not contribute to any observed bias.

The outliers from the thermodynamic comparisons exhibited a different pattern. While fewer profiles overall contained outliers, there tended to be more outliers within these profiles. After analyzing these cases in depth, it was concluded that most of the outliers were likely due to changing atmospheric conditions between measurement times. For example, in the case of the CopterSonde and radiosonde comparison, outliers were found when the CopterSonde and radiosonde traversed temperature and moisture gradients at slightly different times during the morning boundary layer transition. The profile that contained the most outliers, along with some supplementary data from the CLAMPS DL, is shown in Fig. 3.

During this comparison, the radiosonde was launched promptly at 16:00 UTC on July 17, 2018. Due to the very low winds, the radiosonde did not clear the flight area immediately and the CopterSonde flight had to be delayed by two minutes. When the

CopterSonde launched at 16:02 UTC, the radiosonde had already traversed a small inversion at 450 m AGL. The CopterSonde also observed this inversion, though it was at 620 m AGL. The outliers from this profile were in the layer between 450 m and 620 m where the inversion changed height. The likely cause of this rapid change in inversion height was an updraft that formed between 16:00 and 16:06 UTC (Fig. 3b). During this time period, it was noted that a cloud formed while the CopterSonde was
5  ascending. Referencing classical parcel theory, the updraft lifted and cooled the parcel to the lifted condensation level where latent heat release from condensation warmed the environment and modified the lapse rate. This same pattern was observed in multiple thermodynamic profiles that contained outliers and would add bias to the comparisons that originates from different error sources than those being studied here.

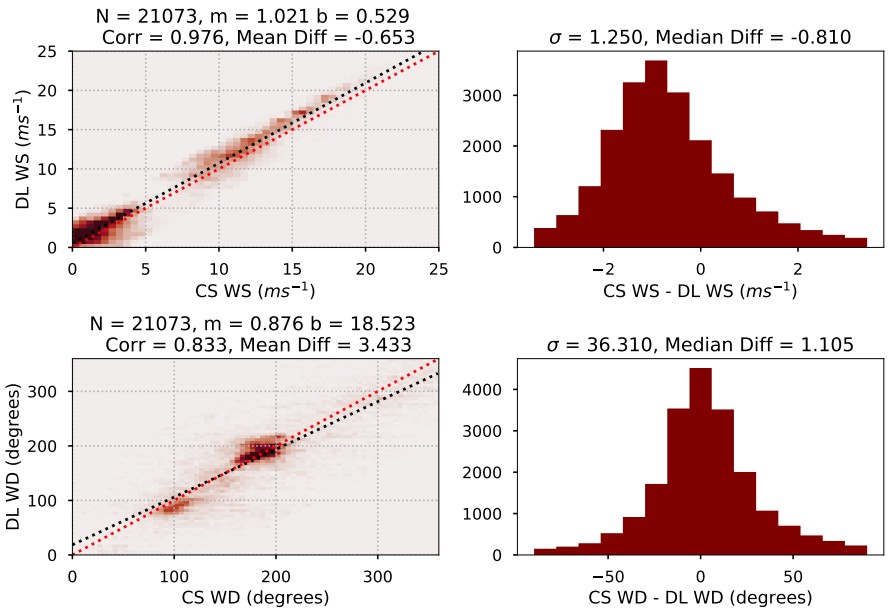

**Figure 4.** Two dimensional histograms of DL measured wind speed vs CopterSonde measured wind speed (a) and DL measured wind direction vs CopterSonde measured wind direction (c). The 2D histograms are binned to $0.5 \text{ m s}^{-1}$ for wind speed and 5 degrees for wind direction. The histograms on the right show the difference in wind speed (b) and wind direction (d). The red dotted line is the 1-to-1 line and the black line is the least-squares regression. The slope (m) and intercept (b) are shown in the title. Various other statistics are also shown in the titles. N corresponds to the number of points, Corr is the Pearson correlation, mean diff is the mean difference between the CopterSonde and the DL, $\sigma$ is the standard deviation of the differences, and median diff is the median difference between the CopterSonde and the DL.

### 4.1.1  CLAMPS vs CopterSonde

10  Kinematic and thermodynamic data from CLAMPS and the CopterSonde are presented in Figs. 4 and 5, respectively. The wind speed and direction from the CopterSonde and the 12-point VAD performed by CLAMPS compare remarkably well. In general, wind speeds less than $5 \text{ m s}^{-1}$ are from LAPSE-RATE while the higher wind speeds are associated with the low-

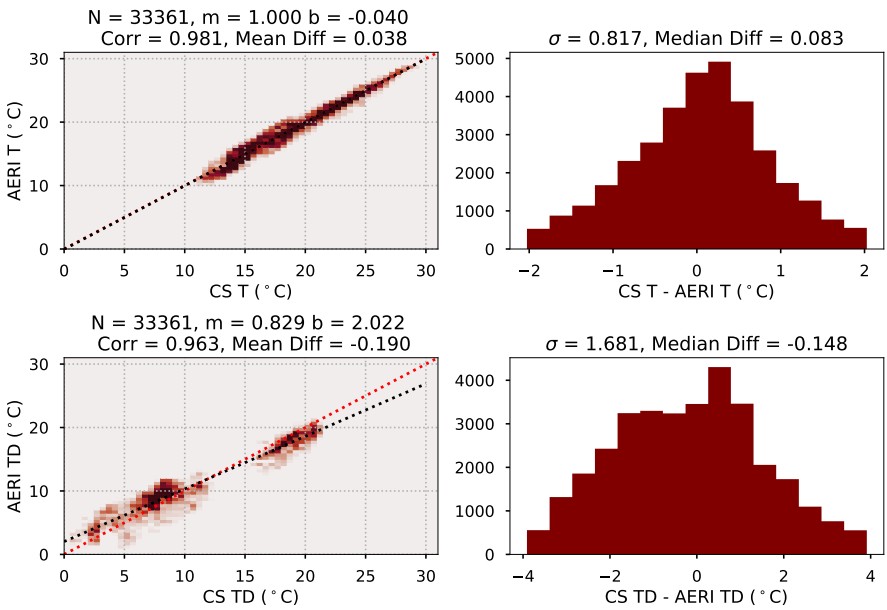

**Figure 5.** Similar to Fig. 4, only for the temperature (a and b) and dew point temperature (c and d). The 2D histograms of temperature and dew point temperature are binned by 0.5°C.

level jet observed during Flux Capacitor. The Pearson correlation (hereafter, just "correlation") between wind speed and wind direction are 0.976 and 0.83, respectively. There is some disagreement in wind speed between the VAD and the CopterSonde, especially at higher wind speeds. This is discussed more in the next section.

The observed wind directions are largely bi-modal (Fig. 4c), with one group of observations around 90 degrees and one group
around 180 degrees. The group around 180 degrees largely consists of observations from the Flux Capacitor dataset (southerly low-level jet) while the group around 90 degree is largely from LAPSE-RATE (easterly katabatic flows from mountains). Additionally, there are points scattered elsewhere that are also largely from LAPSE-RATE.

Overall, the wind directions have a correlation of 0.83. The low wind speeds from LAPSE-RATE make it difficult to accurately measure the wind direction on both the CopterSonde and the CLAMPS DL; the CopterSonde needs strong enough wind
to tilt the craft while the DL needs a strong enough wind to ensure homogeneity over the scan volume. This results in a large standard deviation in the differences. This can also be seen in Fig. 6a. Additionally, there appears to be a small bias in wind directions from LAPSE-RATE. This is likely due to uncertainties in the true heading of the DL.

The comparison of the thermodynamic measurements reveal deficits in the AERI moisture retrieval. The data points for the AERI moisture retrievals have more spread than those for the temperature retrieval, especially in the dry, hot conditions
observed during LAPSE-RATE (dew point temperatures below 13°C in Fig. 5c). The AERI moisture retrievals performed better during the Flux Capacitor campaign (dew point temperatures greater than 15°C), which could be due to the more representative prior used for the retrieval. This prior data is used to help initially constrain the retrievals. The prior dataset used for the Flux

Capacitor retrievals was generated from radiosondes launched from the National Weather Service (NWS) in Norman, OK. The LAPSE-RATE prior was constructed with radiosondes launched by the NWS in Boulder, CO, which may not fully represent the conditions in the San Luis Valley.

Overall, the mean difference in dew point temperature temperature between the CopterSonde and the AERI was -0.190°C with a standard deviation of 1.681°C. The CopterSonde temperature and the AERI temperature retrieval have a high correlation of 0.981. The difference between the CopterSonde and the AERI temperatures average to 0.038°C, with a standard deviation of 0.817°C.

### 4.1.2 Radiosonde vs CopterSonde

Figure 7 shows the wind speeds estimated by the radiosonde compared to the CopterSonde estimate. The wind speeds have a high correlation of 0.969 and a relatively low standard deviation of $1.355 \, \mathrm{m\,s^{-1}}$. The same bias in wind speed presented in Section 4.1.1 is also seen here, indicating the CopterSonde is consistently underestimating the wind speed by approximately $0.75 \, \mathrm{m\,s^{-1}}$. Possible reasons for this bias are discussed in Section 5.

The wind direction from the two systems has a lower correlation (0.853). While the wind directions observed generally agree well (mean difference of 4.204 degrees), there is a large standard deviation (36.854 degrees). Much of this noise results from the low wind speed observations from LAPSE-RATE; both the radiosonde and CopterSonde struggle to capture the correct wind direction when wind speeds are low (Fig. 6b).

The radiosonde and CopterSonde have a high level of agreement between their thermodynamic measurements. The correlation of the temperature and dew point temperature are both 0.99 (Fig. 8). The temperature comparison is slightly better, evidenced by the lower standard deviation (0.408°C) and mean difference (-0.091°C). A bias is observed in dew point temperatures lower than 13°C. This grouping of measurements is entirely from the LAPSE-RATE campaign and there is a consistent offset. Given the CopterSonde was calibrated in a lab setting, while the radiosondes were not, this could be a moist bias on the part of the radiosondes. However, this has not been documented to the knowledge of the authors. It could also be the relative humidity sensors have a pressure dependence. Given the AERI moisture retrievals contained a high amount of spread compared to these instruments, it is difficult to determine which system is causing the bias.

### 4.1.3 Radiosonde vs CLAMPS

Finally, Figs. 9 and 10 show comparisons between radiosondes and CLAMPS. The kinematic measurements from radiosondes and the DL compare well with a correlation of 0.984 for wind speed and 0.899 for wind direction (Fig. 9). There appears to be more noise in the wind directions, corresponding to a mean difference of -11.079 degrees. This is primarily from the low wind speeds observed during LAPSE-RATE where all the systems have difficulty in accurately capturing the wind speed and direction (Fig. 6). There is a slight wind speed bias in one of the instruments, especially at higher wind speeds. However, since the CopterSonde shows bias in Sections 4.1.1 and 4.1.2, it is impossible to determine which instrument has biased measurements. These results are similar to the results of Päschke et al. (2015), in particular the better performance at higher wind speeds.

Figure 10 shows the thermodynamic comparisons between radiosondes and the AERI retrievals. These observations have a high correlation of 0.98 and a mean difference of -0.17°C for temperature. The median difference is 0.249°C and 0.324°C for temperature and dew point temperature, respectively. These results also agree with past comparisons of AERIoe retrievals to radiosondes (Blumberg et al., 2015; Turner and Blumberg, 2018), namely the temperature retrieval tends to perform better (in terms of uncertainty) than the moisture retrieval.

## 4.2 Case Studies

It is meaningful to analyze a couple of case studies in order to better understand how various features observed in the statistical analysis manifest themselves in individual profiles. Case studies also provide a sense of the conditions observed during the campaigns. A representative case from both Flux Capacitor and LAPSE-RATE will be shown to illustrate the different conditions observed.

### 4.2.1 LAPSE-RATE Case Study

The first case considered is for July 19, 2018. During this period, the focus of LAPSE-RATE participants was to capture drainage flows in the northwest part of the valley. CLAMPS and one of the CopterSonde teams continued to operate at the Moffat site during this period. Figure 11 shows the temperatures and wind speeds observed by CLAMPS and the CopterSonde at the Moffat site during this period. Flights started shortly after 11 UTC while there was still a strong nocturnal temperature inversion present and continued until 17 UTC. During this period, some of the highest wind speeds observed during LAPSE-RATE occurred.

Examining an example profile reveals some of the features presented in the statistical analysis. The wind speeds generally all fall within 2 m s$^{-1}$ of each other with the CopterSonde wind speed estimates generally tending to be the lowest (Fig. 12a). In terms of wind direction, the instruments all follow the same general pattern with height and are generally within 20 degrees of each other (Fig.12b). There is a wind speed maximum around 200 m AGL, likely due to the drainage flows from the surrounding mountains. The directional shear layer starting at approximately 600 m AGL also indicates the presence of a slope flow.

The thermodynamic comparison between the AERI, CopterSonde, and radiosonde is shown in Fig. 12c. As would be expected, a nocturnal inversion is present. While all the systems capture the inversion, they are slightly different. The AERI retrieval smooths out the temperature inversion and shows the maximum temperature to be higher both in elevation and temperature than both the UAS and the radiosonde. This is a common occurrence in the data from the LAPSE-RATE campaign and may be due to the prior dataset that was used to generate the initial guess for the LAPSE-RATE retrievals, as mentioned in Section 4.1.1.

Additionally, there is a consistent offset between dew point temperatures measured by the CopterSonde and the radiosondes, which is observed throughout the LAPSE-RATE campaign. This is discussed in more detail in Section 5. In addition, AERI moisture retrieval performs poorly close to the surface. This could be due to a bad surface constraint.

### 4.2.2 Flux Capacitor Case

The next case considered is from the OU organized Flux Capacitor campaign in October 2018 (Fig. 13). During the overnight hours of Flux Capacitor, wind speeds were much higher than the LAPSE-RATE case due to the onset of a nocturnal low-level jet (LLJ, Fig. 13b). This is a common nighttime feature for the Southern Plains, thus it is important to characterize how the CopterSonde performs in these high winds if a 3D Mesonet is to be established.

Figure 14 shows observations from the three systems on October 6, 2018 at 02:32 UTC. During this time period, wind speeds attained a maximum speed of $20 \, \mathrm{m\,s^{-1}}$ around $850 \, \mathrm{m}$ AGL. Due to the high winds, the flight was terminated before reaching $1{,}200 \, \mathrm{m}$; this is one of the limitations of the CopterSonde. The CopterSonde also slightly underestimates the wind speed compared to the other systems once the vertical shear decreases. This could be due to the CopterSonde being calibrated for wind while it is stationary, rather than while it is ascending. It could also simply be that the calibration coefficients for wind speed are not valid for such high velocities. More investigation is needed in this area.

The thermodynamic data from the systems deployed during Flux Capacitor are in better agreement than the data presented in Section 4.2.1. All instruments are able to capture the nocturnal temperature inversion and accurately capture the residual layer. The difference in dew point temperature between the radiosonde and the CopterSonde is much smaller and the instruments agree well. The dew point temperature from the AERI has a slight bias, but captures the general shape of the profile much better.

## 5 Discussion

Overall, the systems tested all perform reasonably well when compared to each other. There are still many nuances to each instrument that need to be taken into account when deciding which system to use operationally. Additionally, some of the systems still need further development in select areas to insure good data quality.

For example, the CopterSonde currently underestimates the wind speed at higher velocities. This could be due to a number of factors. One likely possibility is that coefficients utilized to estimate the wind speed with the tilt of the craft are not valid at higher wind speeds. The coefficients used here were determined while loitering next to a $10 \, \mathrm{m}$ tower at KAEFS while the wind speeds were generally less than $10 \, \mathrm{m\,s^{-1}}$. Additionally, a linear relationship was used to determine these coefficients. As wind speeds increase, the CopterSonde must tilt more to compensate, leading to a larger cross sectional area and thus more drag. This could lead to a non-linear relationship that must be accounted for. Another possible reason for the underestimation is how the profile is performed. As mentioned before, the CopterSonde wind coefficients were determined while hovering at a constant altitude, but for these tests the CopterSonde was ascending at $3 \, \mathrm{m\,s^{-1}}$. This could lead to a discrepancy in the observations since the pitch of the craft is likely different while ascending. Further experiments are being conducted to account for and correct the root cause of the wind speed underestimation.

In terms of thermodynamic measurements, all of the platforms were highly correlated with one another in temperature and dew point temperature. The AERIoe retrievals still need to be refined and there are a number of items that could have contributed to the spread, especially in the moisture fields from LAPSE-RATE. One of the major issues with the AERIoe

retrievals from LAPSE-RATE is the proximity to the a priori dataset used to initially constrain the retrieval. Since there is no long term archive of radiosondes launched in the San Luis Valley, radiosondes launched from the NWS office in Boulder, CO were used, which is over 200 km away. Additionally, uncertainty in the moisture retrievals are inherently larger for the conditions observed during LAPSE-RATE.

Due to these uncertainties in the AERIoe retrievals, it is difficult to determine whether the radiosonde or the CopterSonde exhibited biased moisture measurements during LAPSE-RATE. Since the CopterSonde sensors were well characterized in the Oklahoma Mesonet's calibration laboratory, it seems unlikely the bias could be from the CopterSonde. However, there could be a previously unknown pressure dependence to the sensors being used on the CopterSonde. More work is needed to determine the issue.

This study also highlights synergies between the various systems. For example, Doppler lidars and WxUAS can accurately capture full thermodynamic and kinematic profile at a resolution of 15-30 m, which is better than the specifications laid out by Hoff and Hardesty (2012).

## 6  Conclusions

**Table 2.** Summary of CopterSonde measurement specifications based on the results of this study when compared to the Vaisala RS92-SGP data used in this study

| CopterSonde 2.5 Specifications | |
| --- | --- |
| Temperature | $\pm 0.5^{\circ}$C |
| Dew point temperature | $\pm 0.7^{\circ}$C |
| Horizontal Wind Speed | $\pm 0.6 \, \mathrm{m\,s^{-1}}$ (in speeds $> 4 \, \mathrm{m\,s^{-1}}$) |
| Horizontal Wind Direction | $\pm 4^{\circ}$ (in speeds $> 4 \, \mathrm{m\,s^{-1}}$) |

For meteorologists to fully take advantage of advanced high resolution forecast models, high resolution observations of the
boundary layer are crucial. Two paradigms have been introduced to fill the current observational gap in the boundary layer. In this paper, data from WxUAS (the CopterSonde) and a system of atmospheric profiles (CLAMPS) were compared to the historical profiling standard, the radiosonde, during two different field campaigns.

In this study we found that all the systems agree relatively well, with correlations between all the instruments and variables greater than 0.85, with most above 0.90. The thermodynamic retrievals from CLAMPS perform well in terms of temperature,
though the moisture retrievals could be improved. Additionally, there is still improvement to be made in wind speed estimation on the CopterSonde, mainly related to calibration procedures.

This study presents the most comprehensive comparison of a WxUAS to other profiling systems known to the authors. Additionally, it shows that measurements from WxUAS are comparable to other prominent profiling systems, thus they can provide a low cost alternative to expensive ground-based, remote sensing systems for the 3D Mesonet concept laid out by
Chilson et al. (2019). While the remote sensors and radiosondes presented in this study have been characterized by previous

studies, the CopterSonde is a relatively new platform. Thus we present specifications of the uncertainty in CopterSonde measurements when statistically compared to the RS92-SGP in Table 2. These were determined from the standard deviations of the differences calculated in this study. Note that the wind speed and direction uncertainties were determined using data with wind speeds greater than $4 \, \mathrm{m \, s^{-1}}$.

Remote sensing systems will still have a place where WxUAS may not be able to fly, such as near busy airports or in incredibly remote locations where servicing WxUAS would be difficult. Future work will revolve around eliminating the bias from the CopterSonde wind measurements as well as using UAS to help retrieve other variables from the AERI, such as trace gasses.

*Data availability.*   Data from Flux Capacitor are available upon request to the corresponding author. Data from LAPSE-RATE are publicly
available on the data hosting website Zenodo. The references for each dataset are as follows. CLAMPS Dopper lidar (Bell and Klein, 2020), CLAMPS MWR and surface observations (Bell et al., 2020b), AERIoe retrievals (Bell et al., 2020a), CLAMPS radiosondes (Waugh, 2020), and CopterSonde profiles (Greene et al., 2020)

*Author contributions.*   Conceptualization, T.M.B. and B.R.G.; Methodology, T.M.B., B.R.G., and P.B.C.; Software, T.M.B.; Formal Analysis, T.M.B.; Investigation, T.M.B., B.R.G., P.M.K., M.C., and P.B.C.; Resources, P.M.K, M.C., P.B.C.; Writing - Original Draft, T.M.B.; Writing
- Review & Editing, B.R.G., P.M.K., M.C., and P.B.C.; Visualization, T.M.B; Supervision, P.B.C and P.M.K.; Funding Acquisition, P.B.C. and P.M.K.

*Competing interests.*   The authors declare that they have no conflicts of interest.

*Acknowledgements.*   The authors would like to thank Dr. David Turner for his assistance in getting AERIoe running for the AERI retrievals in this paper. Additionally, the authors would like to thank Dr. Sean Waugh for his assistance in transporting CLAMPS to Colorado for
LAPSE-RATE. This research has been supported in part by the National Science Foundation under Grant No. 1539070 and internal funding from the University of Oklahoma.

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

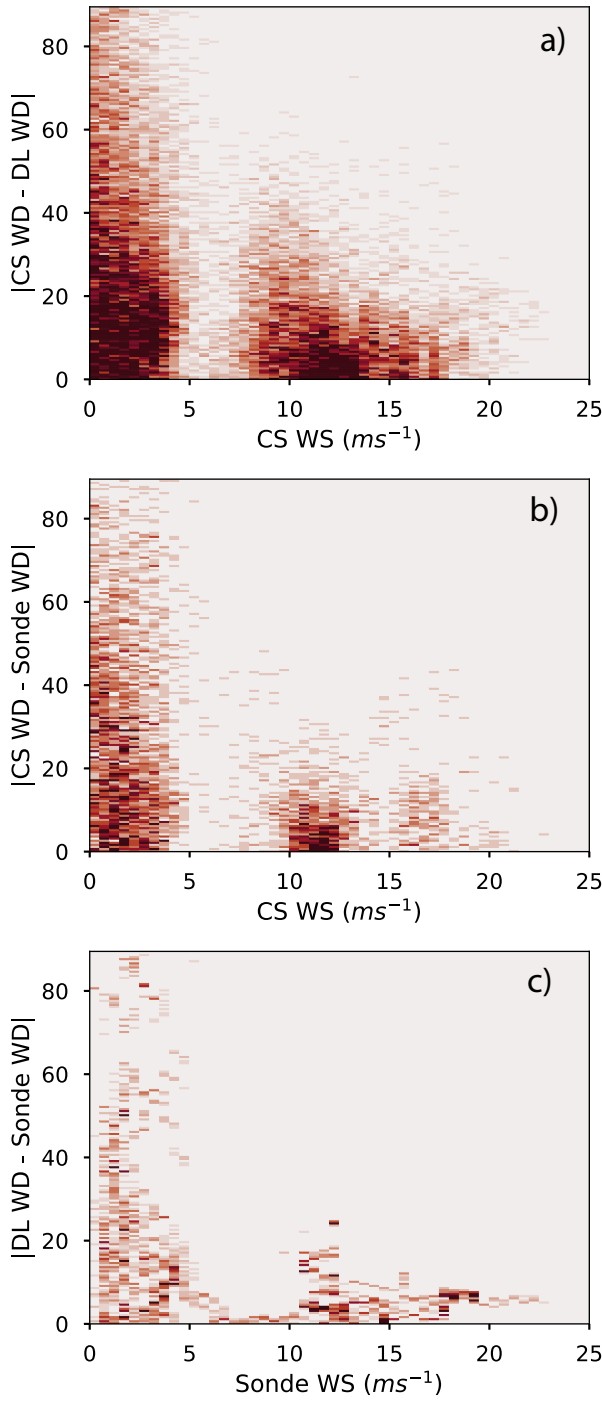

**Figure 6.** Two dimensional histograms of absolute wind direction difference vs wind speed for the CopterSonde and DL (a), CopterSonde and radiosonde (b), and DL and radiosonde (c). This shows that the lower wind speed measurements have a higher level of uncertainty to the wind direction. Again, the distribution is bi-modal, with LAPSE-RATE observations generally all falling below $5\,\mathrm{m\,s^{-1}}$

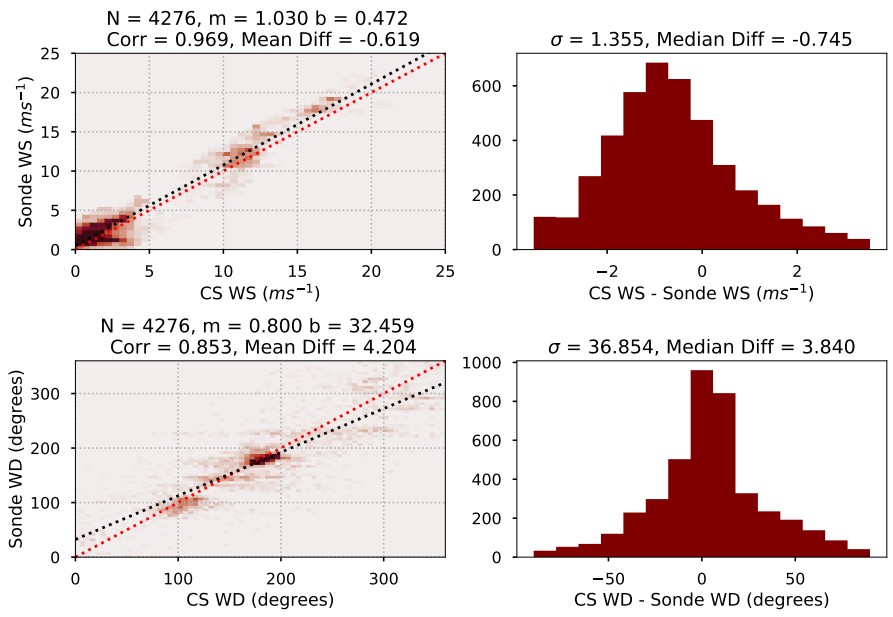

**Figure 7.** Similar to Fig. 4, only comparing kinematic measurements from the radiosondes and the CopterSonde.

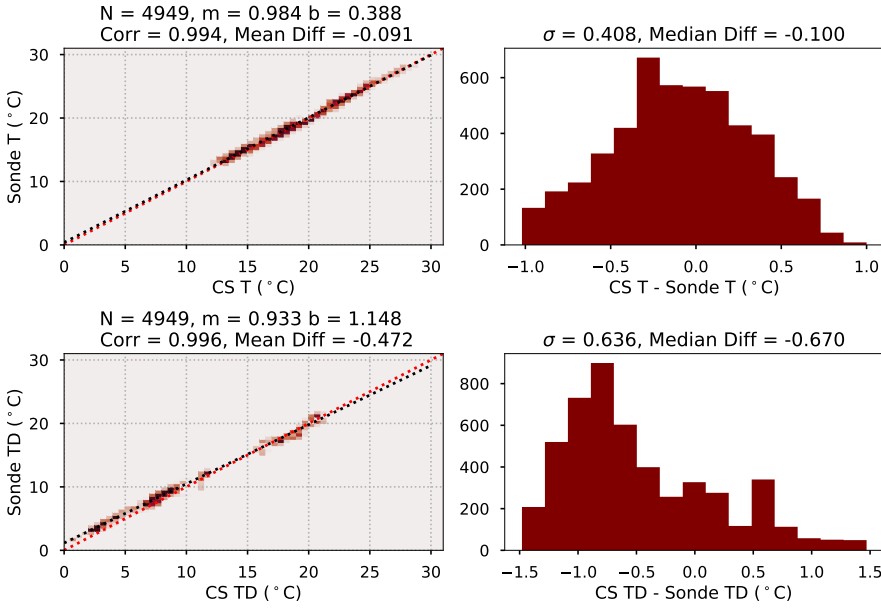

**Figure 8.** Similar to Fig. 5, only comparing measurements from the radiosondes and the CopterSonde.

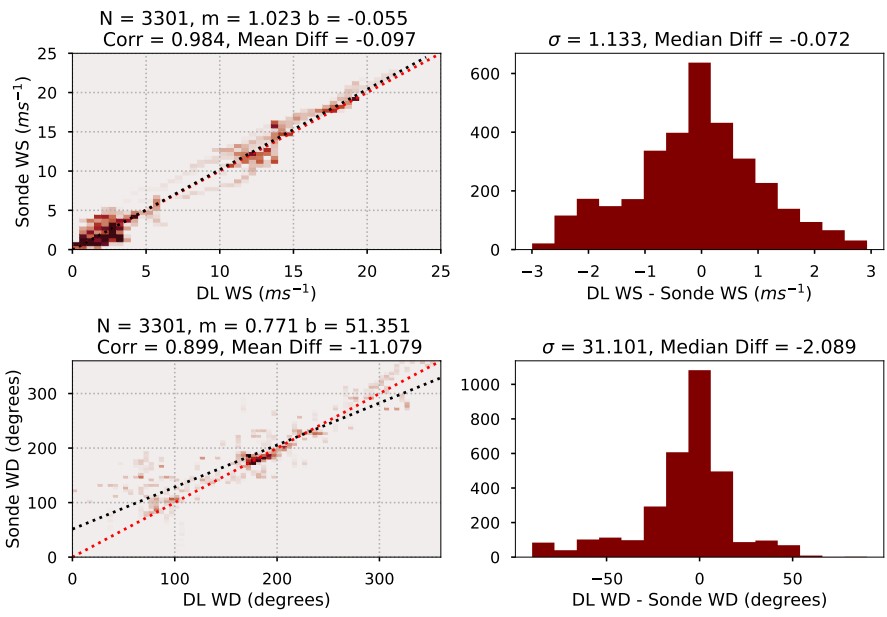

**Figure 9.** Similar to Fig. 4, only comparing kinematic measurements from the radiosondes and the AERI.

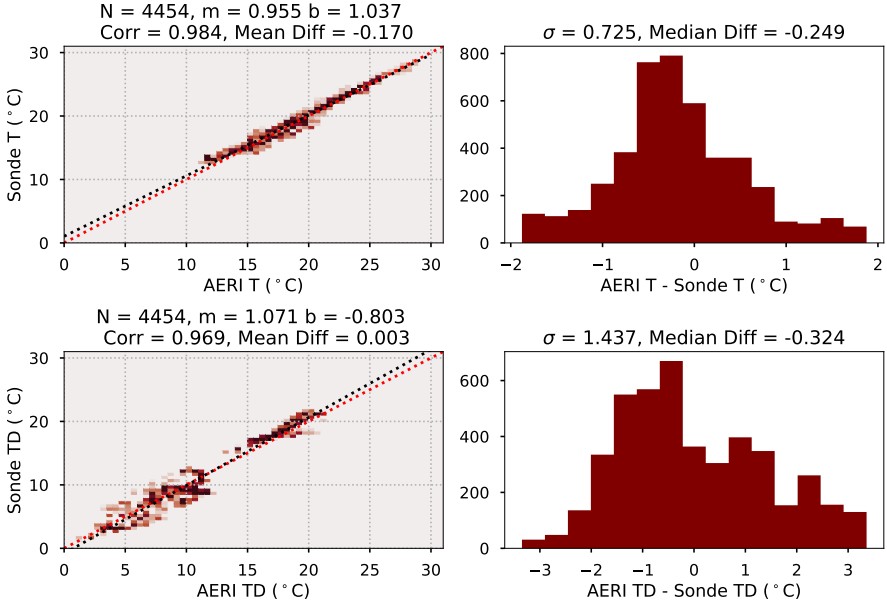

**Figure 10.** Similar to Fig. 5, only comparing thermodynamic measurements from the radiosondes and the DL.

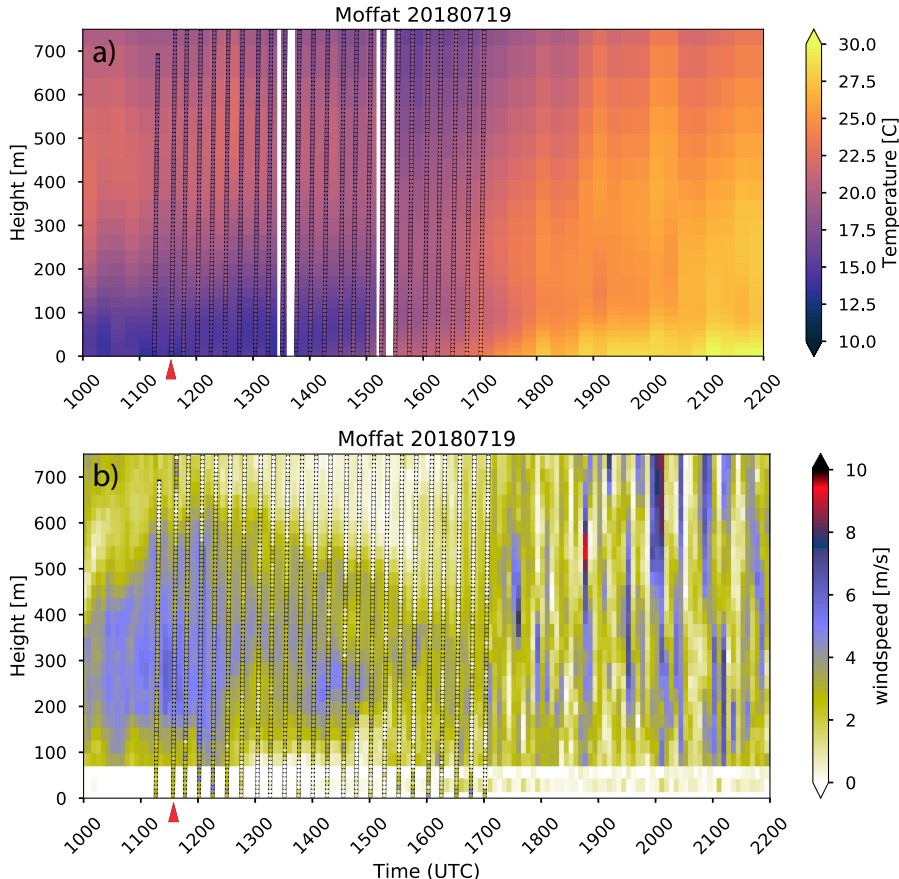

**Figure 11.** Time height plots of temperature (a) and wind speed (b) from the Moffat site on July 19, 2018. The background is temperature from the AERI retrievals while the points overlaid on top are data from the CopterSonde at approximately 9m resolution. The red arrow points to the profile shown in Fig. 12.

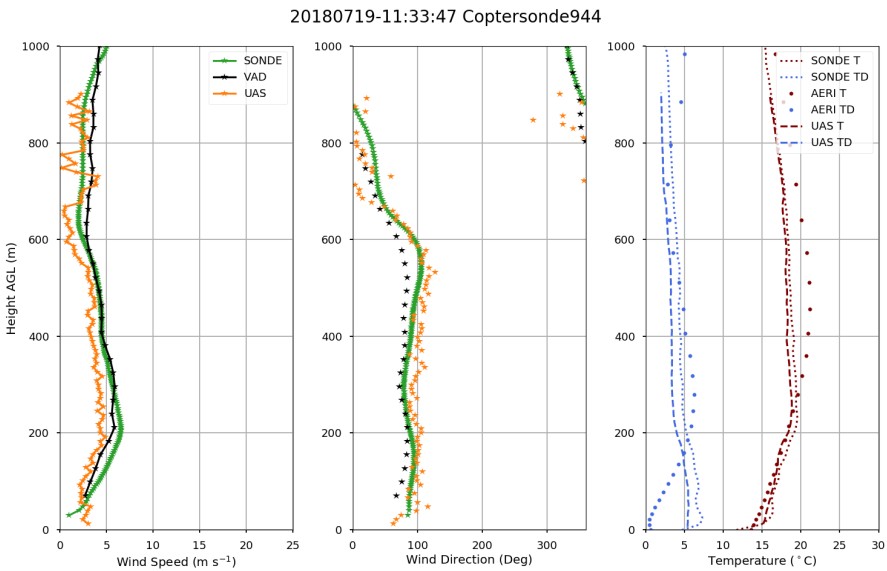

**Figure 12.** Profile plots of wind speed (a), wind direction (b), and temperature and dew point temperature (c) from CLAMPS, the Copter-Sonde, and a radiosonde on July 19, 2018 at 11:33 UTC. The CopterSonde was launched just after the radiosonde, as soon as it was deemed the radiosonde was not in the flight path of the CopterSonde.

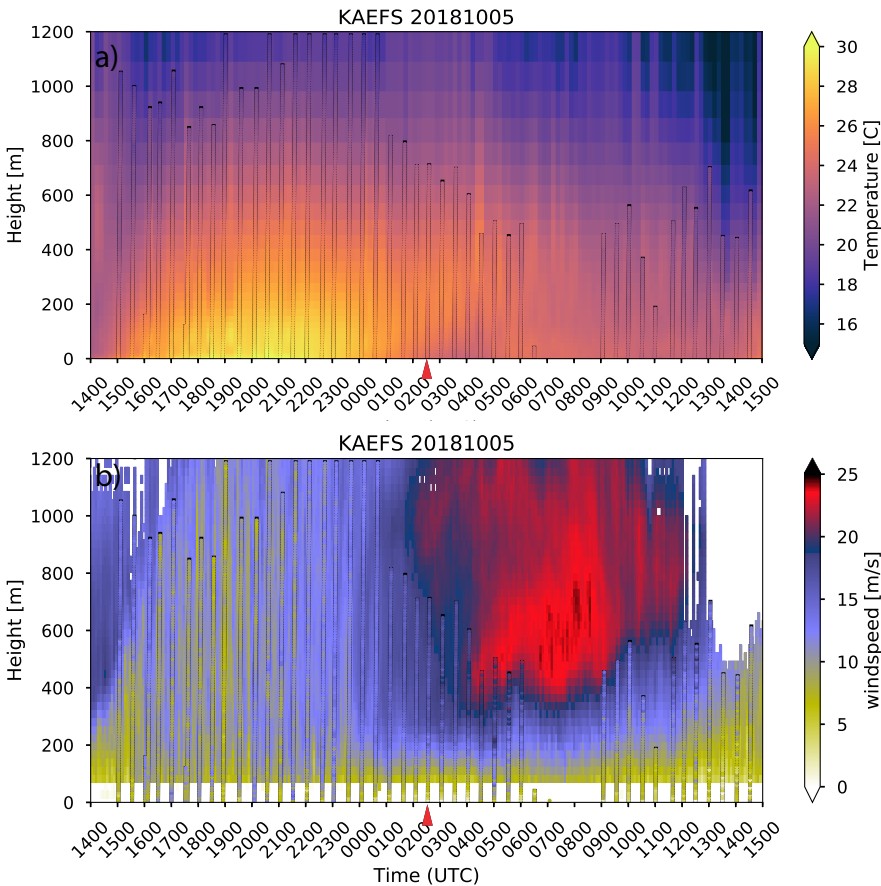

**Figure 13.** Same as Fig. 11, only for October 5-6, 2018 during the Flux Capacitor campaign. These timeheights contain data from the entire observation period.

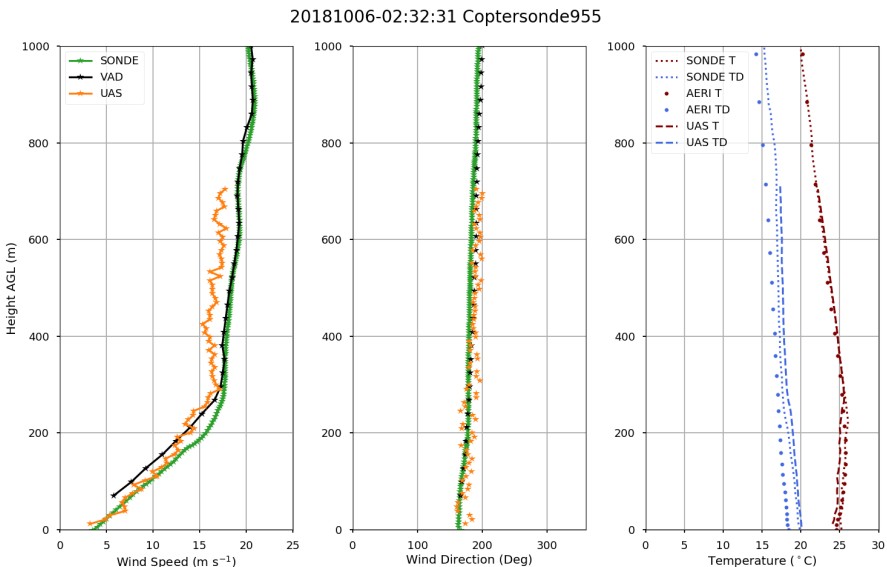

**Figure 14.** Same as Fig. 12, only for October 6, 2018 at KAEFS during Flux Capacitor.