# Peer review of "Confronting the Boundary Layer Data Gap: Evaluating New and Existing Methodologies of Probing the Lower Atmosphere"

_Atmospheric Measurement Techniques, 2019_

## Referee Comment (RC1) · Hubert Luce (Referee) · 16 Jan 2020

**Review of « Confronting the Boundary Layer Data Gap : Evaluating New and Existing Methologies of Probing the Lower Atmospher » by Bell et al.**

by H. Luce (Toulon University)

The manuscript describes results of comparisons between temperature, dew point temperature and wind data collected by radiosondes, a rotary-wing UAV (coptersonde) and a remote-sensing platform (CLAMPS) consisting of several instruments (radiometers, Doppler lidars, …). The purpose is to evaluate the performance of the various systems for operational observations of the boundary layer. To this end, the authors used data collected during two campaigns (LAPSE-RATE, and Flux capacitor).

Having operational systems with known specifications for continuous observations of the boundary layer is indeed an absolute necessity. The development of UAV platforms for atmospheric research is booming and comparisons with more standard systems are also needed. **I thus recommend the publication of this manuscript but after major revisions.** Indeed, there are some issues that must be clarified or perhaps even corrected.

**Major comments** (not ranked in order of importance).

**1)** Greene et al. (2018, 2019) –and previous studies or in review- studied the influence of the rotary-wing UAV environment on the measurements. These studies allowed to design a specific UAV equipment (Coptersonde 2.5) and to improve the mode of operation in order to minimize the instrumental effects on the measurements. On the other hand, the IMet temperature sensor and IST humidity (TU) sensor onboard have been calibrated and possible biases have been corrected.  The authors thus consider the Coptersonde as a "single tool" composed of the drone itself and TU sensors. By doing so, they assume that the TU sensors cannot be changed and they evaluate the performance of the platform as a whole. In their analyses, it is not thus possible separate effects related to the performance of the TU sensors and to UAV technique itself. In order to avoid mixing up the various contributions, it would have been more appropriate to compare the performance of the Coptersonde and balloons equipped with the same TU sensors (radiosondes are equipped with Vaisala sensors). Therefore, the authors should clarify the purpose of the TU comparisons. (This comment applies to TU measurements only. The comparisons between wind measurements are fully relevant because the Coptersonde-derived winds are based on the behavior of the airframe itself.)

**2)** The manuscript does not focus on the newly developed Coptersonde but consider the three platforms on the same level. However, the radiosondes and the Doppler lidar/AERI techniques have been developed well before the UAV technique. Therefore, the comparisons described in section 4.1.3 (Radiosonde vs CLAMPS) are likely not original. They are interesting for the purpose of the manuscript but if earlier studies exist, they should be referred and compared (e.g., Pearson et al., 2009?)

**3)** It is quite unfortunate that the two datasets of wind and humidity from LAPSE-RATE and Flux Capacitor are so different. It is then difficult to know if the estimated performance is reproducible or not. For example, the difference between the wind speed measured by the radiosondes and the Coptersonde is weak during LAPSE-RATE when the wind is low (<~5 m/s) and is "large" when the wind is large (>10 m/s) during Flux Capacitor (Fig. 5). Therefore, we do not know if a bias occurs only because the wind is large or if the fact that the data have been collected during different campaigns is important or not. In addition to the available plots, plotting the results from LAPSE-RATE and Flux Capacitor separately might be informative.  Also, although the largest differences in direction are very likely due to low wind conditions as the authors stated, an additional panel in Fig 3, 5 and 7 showing differences

of wind direction WD vs the (mean) wind speed WS would give additional credence to this assertion. This figure would also give an indication of the lower threshold on WS from which the wind directions are "reasonably" consistent. Note that the wind direction uncertainty of 2° (page 5, line 17) specified by Vaisala for RS92SGP radiosondes is valid for WS>3 m/s only. It is likely significantly larger for WS<3 m/s.

The x and y labels of the left panels of Fig 3, 5 and 7 seem to be reversed. Please correct.

**4)** Page 8, line 15, it is indicated that "data points with a difference that lie outside the $2\sigma$ envelope are considered outliers and were removed from the analysis". It is likely reasonable to do so, but this aspect should be expanded, considering its importance for the purpose of the manuscript ("evaluation of performance"). In particular, what are the possible causes of these outliers? are all of them identified? what platform is the most contaminated? Very importantly, how many data points (percentage) have been removed?

**5)** As in previous studies reported by the same group, the dew point temperature $T_d$ is shown for quantifying the performance of the humidity measurements. However, I wonder if this parameter is really appropriate for the present purpose, since $T_d$ is not measured but estimated from both measured relative humidity RH and T. Therefore, $T_d$ does not contain possible biases due to humidity measurements only but also from temperature measurements. Why not showing RH?

**6)** It would be more appropriate to show the case studies (4.2) first and then the statistics (4.1). Indeed, the statistics generalize, to some extent, the tendencies shown by the case studies. In Figure 10 and 12, the best agreement is obtained between Doppler lidar and radiosonde WS profiles, confirming Pearson et al. (2009). A careful scrutiny of the radiosonde profile seems to indicate a slight (negative) vertical offset with respect to the 2 other profiles. Since the profiles start from 0 m AGL, I suspect that the radiosonde profiles have not been corrected from the length of the rope between the payload and the balloon. Indeed, the wind measured by the radiosonde is not the wind at the altitude of the radiosonde but the wind at the altitude of the balloon. The altitude difference is only a few meters when launching but it should quickly reach 30 m (?) when the rope is totally unwound. 30 m is the length of the rope recommended by Vaisala for a ~300-g rubber balloon, but the authors should apply an altitude correction (+L m) corresponding to the expected length L of the rope once unwound. 30 m is not negligible here and would provide even better agreements (at least in Figure 10 and 12).

**7)** I do not understand how Table 2 was obtained and what the values mean. How did the authors produce the values indicated in this table from the results described in the manuscript? In particular, pressure and RH data have not been discussed at all.

**Minor comments:**

**1)** The abstract and paragraph 1 of introduction should clearly indicate what atmospheric parameters are compared. The accuracies are discussed in paragraph 2 of introduction without introducing the measurable quantities first.

**2)** In paragraph 2 of Introduction, the advantages and drawback of the WV-DIAL should be shortly described for the sake of completeness.

**3)** The NRC report recommended accuracies and vertical resolution, but what about time resolution?

**4)** Page 2, lines 25-30: the references to fixed-wing aircrafts do not include the DataHawk aircraft developed by Profs. Lawrence and Balsley. Lawrence and Balsley (2013,

*https://doi.org/10.1175/JTECH-D-12-00089.1*, Balsley et al. 2013, DOI 10.1007/s10546-012-9774-x), among others, developed for turbulence observations in addition to standard PTU, wind parameters.

**5)** Page 3, section 2: first line, please define the acronym here.

**6)** Page 4, line 2: Please indicate that the Humidity and Temperature profiler is a microwave radiometer (?).

**7)** Page 5, line 3: please explain why "three" (?) RH sensors.

**8)** Page 5, line 16: The authors mentioned that the radiosonde data are collected every 1 sec. It is true for the raw data for which (GPS) altitudes are not available. The GPS altitude is given in the processed files given at a time sampling of 2 sec. Can the authors clarify what data were used?

**9)** Table 1: please indicate what "average $\Delta t$" means.

**10)** Figures 9 and 11: the data representation is nice but is it possible to reduce the linewidth of the black lines (for more legibility)?

---

## Referee Comment (RC2) · Anonymous Referee #2 · 1 Mar 2020

Review of "Confronting the Boundary Layer Data Gap: Evaluating New and Existing Methodologies of Probing the Lower Atmosphere" By Tyler M. Bell, Brian R. Greene, Petra M. Klein, Matthew Carney, and Phillip B. Chilson The accurate sounding of the atmospheric surface layer and boundary layer with good vertical resolution remains a challenge for the lower atmospheric research community. The traditional measurement techniques like radiosondes, microwave radiometers, lidar often inadequate to produce desired boundary layer sounding data. In this paper, the authors present an improved version of the CopterSonde used for the boundary layer soundings. Additionally, a systematic and detailed comparison study of CopterSonde data with the data collected using the traditional sounding systems provides the strength and weaknesses of the current approach that utilizes the in situ and remote sensing measurements. The paper is well written and timely. I recommend the manuscript for publication in the Atmospheric Measurement Techniques. Comments 1. Please include a table of CopterSonde sensor details, such as accuracy and range. 2. Getting insitu profiles of surface layer over the oceans is very difficult than the land-based observations. I wonder whether the CopterSonde conducted observations over the ocean? If yes, please include an analysis of the data in the current manuscript. 3. Figures 9 and 11: These figures needs redo to get a better picture of the vertical variabilities.

---

## Author Comment (AC1) · 29 Mar 2020

We thank Reviewer #2 for providing their review on our manuscript. The attached .zip archive contains our response and, in a separate file, a latexdiff version of our revised manuscript.

Please also note the supplement to this comment: https://www.atmos-meas-tech-discuss.net/amt-2019-453/amt-2019-453-AC1-supplement.zip

---

## Referee Report (RR1)

**2ⁿᵈ Review of « Confronting the Boundary Layer Data Gap : Evaluating New and Existing Methologies of Probing the Lower Atmospher » by Bell et al.**

by H. Luce (Toulon University)

After some necessary corrections, the authors improved the manuscript. I have still two important comments to make (that do not prevent its acceptation).

(1) **About the purpose of the manuscript** (comment 1). If the objective is to evaluate the performance of the Coptersonde with the available sensors onboard (IMET and IST sensors) for operational use, it makes sense. If not, considering the equipped Coptersonde as a whole, without clarifying the contributions of the sensors and Coptersonde separately provides lower value to this study. I think that the response given by the authors below should be rephrased and included in the text.

> *You are correct that the Greene et al. (2018, 2019) studies have worked to mitigate the effect of the UAS on the TU measurements, which would decouple the UAS technique from the sensors themselves. However, the recommendations made from these studies were done with fairly idealized setups. The purpose of continuing to evaluate the CopterSonde as a whole is to verify the results from the idealized situations while in a more "operational" mode, especially against a radiosonde that is as widely used as the RS92SGP.*

(2) **About outliers**. In statistics, an outlier is a data point that differs significantly from other observations. Thus, in principle, they may have their own distribution. Considering data points associated with the tail of a distribution as outliers is not necessarily true. They can be outliers or not because the threshold is necessarily arbitrary. Rejecting doubtful data when they differ two much is necessary for scientific works (ie, when physical processes are studied), but is questionable when technical performance is studied. These doubtful data (or, at least, some of them) are part of the dataset to analyze. If their number is very small, if their origin is well identified and if they are expected not to occur again in operational use, then they can be ignored (but, it is the case here?). Otherwise, their characteristics should be specified. With the rejection method applied by the authors, it is not possible to know if the rejected data occur sporadically or if it is a whole "package" of consecutive points. The consequences of the rejection process would not be the same.

I understand that this topic is not easy to tackle, but I think that it should be expanded a little bit by including some examples of rejected data in profiles and scatter plots and discussing the impacts of these rejected data on the statistics (e.g do they affect or not the mean differences, do they introduce biases, etc..)?

**Minor comments:**

**1)** In abstract, (and elsewhere in the text) "dew point"-> "dew point temperature"

**2)** Page 2, lines 30: the references Lawrence and Balsley (2013, _https://doi.org/10.1175/JTECH-D-12-00089.1_, Balsley et al. 2013, DOI 10.1007/s10546-012-9774-x) are in the reference list, but not in the text.

**3)** Page 7 table 1, caption: "flights.xs" ?

---

## Author Response (AR2)

**Author Response to Reviewer**

After some necessary corrections, the authors improved the manuscript. I have still two important comments to make (that do not prevent its acceptation).

*The authors thank the reviewer for their further comments on the manuscript. Each comment is addressed individually below.*

(1) About the purpose of the manuscript (comment 1). If the objective is to evaluate the performance of the Coptersonde with the available sensors onboard (IMET and IST sensors) for operational use, it makes sense. If not, considering the equipped Coptersonde as a whole, without clarifying the contributions of the sensors and Coptersonde separately provides lower value to this study. I think that the response given by the authors below should be rephrased and included in the text.

*To further clarify the purpose of the manuscript, the response we provided previously has been reworded and included in the text, as suggested. The text can be found on Page 3 lines 20-24.*

(2) About outliers. In statistics, an outlier is a data point that differs significantly from other observations. Thus, in principle, they may have their own distribution. Considering data points associated with the tail of a distribution as outliers is not necessarily true. They can be outliers or not because the threshold is necessarily arbitrary. Rejecting doubtful data when they differ two much is necessary for scientific works (ie, when physical processes are studied), but is questionable when technical performance is studied. These doubtful data (or, at least, some of them) are part of the dataset to analyze. If their number is very small, if their origin is well identified and if they are expected not to occur again in operational use, then they can be ignored (but, it is the case here?). Otherwise, their characteristics should be specified. With the rejection method applied by the authors, it is not possible to know if the rejected data occur sporadically or if it is a whole "package" of consecutive points. The consequences of the rejection process would not be the same.

I understand that this topic is not easy to tackle, but I think that it should be expanded a little bit by including some examples of rejected data in profiles and scatter plots and discussing the impacts of these rejected data on the statistics (e.g do they affect or not the mean differences, do they introduce biases, etc..)?

*The authors agree that looking at the outliers is an important aspect, as they are often the most interesting points to look at. The goal of the outlier removal was to eliminate points where a direct comparison due to changing atmospheric conditions was not valid.*

*We carefully re-examined the outliers and looked for any patterns with respect to height, time of day, atmospheric conditions, etc. We found no discernable pattern in the kinematic comparisons; the outliers were random points that could have been a wind gust or random spike that made it through the quality control measures. However, the thermodynamic comparisons did exhibit more of a pattern. Specifically, the profiles with the most outliers all were when conditions were changing rapidly and the temporal/spatial separation of the instruments was a factor. To illustrate this, we included an examination of the profile with the most outliers (see the beginning of section 4.1). This actually proved to be a very interesting case study since a cloud formed while the CopterSonde and radiosonde were ascending. This happened a handful of times during LAPSE-RATE and the following 2 profiles with most outliers all exhibited a similar pattern.*

Minor comments:

1) In abstract, (and elsewhere in the text) "dew point"-> "dew point temperature"

*This has been addressed.*

2) Page 2, lines 30: the references Lawrence and Balsley (2013, https://doi.org/10.1175/JTECH-D-12- 00089.1, Balsley et al. 2013, DOI 10.1007/s10546-012-9774-x) are in the reference list, but not in the text.

*This looks to have been a BibTex error that was not caught. This has been addressed*

3) Page 7 table 1, caption: "flights.xs" ?

*This has been addressed*

[revised manuscript text omitted]